# Effect of COVID-19 Non-Pharmaceutical Interventions and the Implications for Human Rights

**DOI:** 10.3390/ijerph18010217

**Published:** 2020-12-30

**Authors:** Seung-Hun Hong, Ha Hwang, Min-Hye Park

**Affiliations:** 1Division of Regulatory Innovation Research, Korea Institute of Public Administration, Seoul 03367, Korea; Seunghun.hong@kipa.re.kr; 2Division of Disaster and Safety Research, Korea Institute of Public Administration, Seoul 03367, Korea; 3Mechanical and Aerospace Engineering, Ulsan National Institute of Science and Technology, Ulsan 44919, Korea; rabomh@gmail.com

**Keywords:** COVID-19, non-pharmaceutical interventions (NPIs), human rights, effectiveness, contact tracing, school closures

## Abstract

In response to the COVID-19 pandemic, many governments swiftly decided to order nationwide lockdowns based on limited evidence that such extreme measures were effective in containing the epidemic. A growing concern is that governments were given little time to adopt effective and proportional interventions protecting citizens’ lives while observing their freedom and rights. This paper examines the effectiveness of non-pharmaceutical interventions (NPIs) in containing COVID-19, by conducting a linear regression over 108 countries, and the implication for human rights. The regression results are supported by evidence that shows the change in 10 selected countries’ responding strategies and their effects as the confirmed cases increase. We found that school closures are effective in containing COVID-19 only when they are implemented along with complete contact tracing. Our findings imply that to contain COVID-19 effectively and minimize the risk of human rights abuses, governments should consider implementing prudently designed full contact tracing and school closure policies, among others. Minimizing the risk of human rights abuses should be a principle even when full contact tracing is implemented.

## 1. Introduction

In his eloquent explanation of Panopticism, Foucault described how a 17th century European state hit by plague could shut down local communities and carry out surveillance on every corner of the street to oppress people’s movement and assembly [1]. What happened four centuries later was not so different. Faced with an uncontrollable epidemic, even democratic governments swiftly decided to order nationwide lockdowns based on limited evidence that such an extreme measure was effective in containing the epidemic. Indeed, the COVID-19 epidemic proceeded to a global pandemic with unprecedented pace and scope, allowing governments little time to adopt effective and proportional interventions protecting citizens’ lives while observing their freedom and rights [2]. COVID-19 has worldwide infected 77 million people and killed over 1.7 million as of December 2020 [3], becoming the biggest cause of death among infectious diseases, outnumbering tuberculosis [4].

The burgeoning literature examines the effect of non-pharmaceutical interventions (NPIs) [5,6,7,8,9], a state’s capacity to respond to the pandemic [10,11,12], or factors affecting compliance with NPIs [13,14]. Studies seem to reach a consensus that lockdown is effective in containing COVID-19, but are only based on single-country studies [5,8] and do not even explain what lockdown means in different countries [7,15,16]. Given the variety of scope and measures included in lockdowns in different countries, however, it is crucial to investigate the effect of NPIs in a more detailed way. The Oxford COVID-19 Government Response Tracker (OxCGRT) database [17], which tracks 12 COVID-19 government responses over 178 countries, indicates that only 4 countries have implemented complete lockdowns, i.e., a combination of all assembly and movement restrictions. 

This paper examines the effectiveness of NPIs in containing COVID-19 by conducting linear regression over 108 countries. The analysis covers the period from 1 January until 15 June 2020, when some countries began to relax the level of intervention. For a more nuanced analysis, we classify NPIs into three categories according to which NPIs put restrictions on human activities: the restriction on assembly, movement, and privacy. This categorization enables us to assess the variation of government approaches to containing COVID-19 and evaluate the effect of such approaches on curbing the transmission. Then we support the linear regression results by presenting evidence that shows the change in 10 selected countries’ responding strategies and their effects as the confirmed cases increase. We select five from North America and Europe and another five from Asia and the Pacific to visualize the cumulative number of confirmed cases and the change in the NPI levels over time. 

The remainder of the paper comprises three sections. Section 2 introduces research materials and methods. Section 3 presents the results of multiple linear regression and the effects of 10 selected countries’ response strategies. Section 4 discusses the main findings from the empirical studies. Section 5 concludes with lessons learnt and limits of this study. 

## 2. Materials and Methods 

### 2.1. Non-Pharmaceutical Interventions and Their Restrictions 

For examining the effect of NPIs, we selected nine NPIs in the OxCGRT database and broke them down into three categories according to the type of restrictions such responses had on human activities. It is believed that COVID-19 spreads through close human-to-human contact, though it is yet to be confirmed at the time of writing how the virus is transmitted from an infectious person to those they infect [18,19,20,21]. NPIs are mainly designed to curb virus transmission through direct human-to-human contact in two ways: by restricting people’s movement and gathering. One purpose of NPIs is to limit people’s movement. Stay at home requirements, for example, are intended to get both infected and uninfected people not to move around so that they stay away from potential virus transmission. Other NPIs restricting people’s movement include travel bans, such as public transport closures, domestic travel restrictions, and international travel bans, which are all intended to mitigate, among others, the risk of virus transmission caused by human movement. We call NPIs of this sort Movement Restrictions.

Some studies assume that NPIs are purported only to restrain people’s mobility [22]. However, NPIs may also put restrictions on human gatherings. This type of NPI aims at mitigating the risk of virus transmission caused by people’s gathering in a place rather than transmission caused by people’s moving from one location to another. NPIs of this sort include school closures, workplace closures, public event cancellations, and gathering size restrictions. We classify these NPIs as Assembly Restrictions. 

In addition to the two types of restrictions on human-to-human contact, we also consider contact tracing (that is, tracking down personal contacts of all (or selected) confirmed cases) as a third type of restriction for curbing virus transmission. The goal of contact tracing is to trace virus transmission via direct human contact to find out those who are exposed to virus infection. However, its side effects remain that tracking personal contacts may give rise to too much state surveillance over the private sphere. Some countries were less inclined to adopt full contact tracing policies due to rising concerns over privacy violation [23,24]. Thus, we call contact tracing a Privacy Restriction. Table 1 summarizes the categorization of NPIs used in this study.

### 2.2. Decrease Rate of Increase in Cumulative COVID-19 Confirmed Cases 

Several previous studies used the number of daily confirmed cases or deaths as explained variables to identify the effect of NPIs on containing COVID19 [6,7,8,9,17]. However, daily confirmed cases or daily number of deaths are not suitable for quantitatively estimating the effectiveness of NPIs because the daily deviation is large, and the effects do not appear immediately after implementing these policies due to the incubation period of the virus. 

As an alternative, we calculated the rate of increase or decrease in the cumulated number of confirmed cases (DRICs) of COVID-19. To generate this variable, the highest value of the average increase rate of the cumulated number of confirmed cases (IRCs) between the 1 January and the 15 June 2020 was divided by the average IRCs six days after the record date for each country, considering the incubation period [6]. The more the IRCs decreases during the incubation period, the greater the value of DRICs. We used this value as the dependent variable to estimate the effect of NPIs applied at the time when the IRCs was highest. Of course, there are many cases where the dates the highest IRCs were recorded and the dates when associated NPIs were implemented do not coincide. Considering that the incubation period of COVID-19 is about six days, however, it is an appropriate assumption. A total of 108 countries were included in the linear regression analyses except for countries in which the cumulated number of confirmed case graphs do not show a sigmoid curve or calculation of DRICs is impossible due to missing data or errors.

The model we employed is basically cross-sectional, but to consider the incubation period in which the effect of NPIs is manifested, DRICs, the dependent variable with a longitudinal character is devised. To define DRICs, we first define ${\bar{y}}_{t}$ as the mean of the cumulative confirmed cases for three days to make the cumulative confirmed case curve smooth (see Equation (1)).
(1)$${\bar{y}}_{t}=\frac{y_{t-1}+y_{t}+y_{t+1}}{3}$$
where *y_t_* = cumulative confirmed cases of COVID-19 on day *t* and *t* = number of days from the 1 January 2020.

Second, we define a_t_ as the average rate of increase in confirmed cases over a week (see Figure 1 and Equation (2)).
(2)$$a_{t}=\frac{\Delta \bar{y}}{\Delta t}=\frac{{\bar{y}}_{t+3}-{\bar{y}}_{t-3}}{6}$$

Finally, we define DRICs (Φ) (see Figure 1 and Equation (3)).
(3)$$\Phi =\frac{a_{t^{\prime }}}{a_{t^{\prime }+6}}$$
where *t*’ = the day (*t*) when *a_t_*_’_ = max (*a_t_*).

### 2.3. Multiple Linear Regression Models 

We established multiple linear regression models with DRICs as the dependent variables and NPIs as the independent variables to confirm the effect of NPIs on containing COVID-19 (see Equation (4)).
(4)$$\Phi =\underset{i}{\sum }\beta_{i}\cdot NPI_{i}+\varepsilon$$

The standardization of NPIs relies on the method suggested by OxCGRT [17], considering description 1 (degree) and description 2 (targeted or general) (see Equation (5)). One difference is that the scale of the variables is normalized from 0 to 1.
(5)$$NPI_{j,t{}^{\prime }}=\frac{v_{j,t{}^{\prime }}-0.5\left(only\;if\;description\;2=targeted\right)}{\max\left(v_{j}\right)}$$
where NPI = standardized NPI score; *j* = kind of *NPIs* (A1 to A4, M1 to M4, and P1); and *v_j,t_*_’_ = the value of description 1 of NPI *j* on day *t*’.

### 2.4. Ten Countries’ Response Strategies

In order to confirm the results of the multiple linear regression analysis, we examined the changes in response strategies over time and the changes in the number of cumulative confirmed cases in each country. To present more explicit graphs, we combined NPIs for Assembly Restrictions and Movement Restrictions into one indicator, each using the method suggested by OxCGRT [17] (see Equations (6a) to (6c)).
(6a)$$NPI_{\bar{A},t}=\frac{1}{4}\left(I_{A1,t}+I_{A2,t}+I_{A3,t}+I_{A4,t}\right)$$
(6b)$$NPI_{\bar{M},t}=\frac{1}{4}\left(I_{M1,t}+I_{M2,t}+I_{M3,t}+I_{M4,t}\right)$$
(6c)$$NPI_{\bar{P},t}=I_{P1,t}$$

## 3. Result

### 3.1. Descriptive Statistics

The average value of DRICs is about 1.599, which indicates that the countries used in the analysis showed a decrease in IRCs to the 62.5% (1/1.599) level on average after six days when the IRCs were recorded as the highest. The Shapiro–Wilcoxon test confirms the normality of the dependent variable (W = 0.69185, *p*-value < 0.0001).

Table 1 shows details of the explanatory variables and Table 2 displays descriptive statistics. School closures are the most widely and strongly implemented NPI, followed by canceling public events. 

The correlation matrix of the dependent and independent variables used in the regression models shows that only school closures and contact tracing interventions are positively correlated with DRICs while others show negative correlations (see Table 3). Among the independent variables, stay at home requirements and internal movement restrictions are highly correlated (r = 0.71), and other variables show moderate or low correlations. However, the highest score of the variance inflation factor (VIF) is 2.42, which suggests very weak evidence of multicollinearity.

### 3.2. Multiple Linear Regression Analysis

Table 4 displays the results of the multiple linear regression models. According to the results of Model 1, school closure has a strong, positive impact on containing COVID-19, while the rest of the NPIs are statistically insignificant or have negative effects. These conflicting results may draw on the unpredictability of people’s behaviors when their movement and assembly are partially limited. When restricting people’s gathering or movement, a government expects them to stay in their own homes. The negative correlation between most NPIs except school closures and DRICs implies that such measures, when implemented individually, would increase other communal activities. A study using Google’s mobility data revealed that people’s mobility has diversified since the outbreak of COVID-19, rather than having been restrained [22].

On the contrary, school closures are effective because, even though schools are physically closed, school terms still carry on and students are obliged to fulfill their distance learning at home. At school, students have to stay in the same, enclosed place for hours, maintaining contact with each other. This would not happen in distance learning, where students are asked to stay at home and not mix with one another. Furthermore, as younger students stay at home to take online classes, at least one parent must look after them, which may result in fewer outdoor activities and more compliance with stay at home policies.

Model 2 and 3 show the effects of the sequential combination of contact tracing, a Privacy Restriction, with variables related to Assembly and Movement Restrictions. Unlike NPIs which directly limit people’s assembly or movement to lower the probability of infection throughout the community, contact tracing is a proactive intervention to prevent further spread by individually examining people with a high probability of infection, i.e., whether or not implementing contact tracing can affect the impact of other NPIs. Under this assumption, we used contact tracing as the interaction term. Model 2, which includes variables that combine contact tracing and Assembly Restrictions, shows an 11.5% increase in explanatory power (Adj. R-squared = 0.343) than Model 1 (Adj. R-squared = 0.228), and it turns out that 3 out of 4 combined variables were statistically significant. On the other hand, Model 3 shows only a 0.3% increase in explanatory power (Adj. R-squared = 0.346) compared to Model 2, and there was no additional variable that shows statistical significance.

According to the results of Model 2, the effect of school closing is (−2.070 + 4.162×P1). Therefore, a complete contact tracing policy (P1 = 1) must be applied together to have the effect of containing COVID-19 through school closing. The effect of contact tracing is (4.162*A1−1.221*A2-3.163*A3). Unlike school closing, workplace closing and canceling public events appear to reduce the effectiveness of contact tracing. One reason may be that limiting predictable gatherings may push people to have unpredictable gatherings and movement behaviors rather than having them stay at home [5]. 

We found that school closing is effective in containing COVID-19 only when it is implemented along with complete contact tracing. Then, our next question was whether there is a clear difference in the degree of containing COVID-19 between countries adopting limited contact tracing and those implementing complete contact tracing. The DRIC scores of countries with limited contact tracing (*n* = 47) and those with complete contact tracing (n = 46) are 1.372 and 1.795, respectively, and the difference between the two scores is statistically significant (t = −3.166, df = 54.031, *p*-value = 0.003). The IRCs of countries with limited contact tracing decreased to about 72.9% (1/1.372) after six days from the highest, while those with complete contact tracing decreased to 55.7% (1/1.795). This result shows that a stronger contact tracing policy is more effective in containing COVID-19.

### 3.3. Ten Countries’ Response Strategies and Their Effects

The results of our linear regression analysis are meaningful in that they correlate the effect of governments’ NPIs by country in the period when confirmed cases increased the most. Still, there are limitations in showing the change in individual countries’ response strategies and their effects. Therefore, we confirm the results of the regression analysis by examining the changes in the cumulated number of confirmed cases and the changes in government policies of ten selected countries. Five countries are selected from North America and Europe, and another five from Asia and the Pacific (see Figure 2). 

Figure 2 demonstrates that in most countries, the stringency of Assembly and Movement Restrictions surges in response to the increase of newly confirmed cases. A striking difference is that countries which implemented complete contact tracing from a very early stage of the COVID-19 epidemic, such as China, South Korea, Vietnam, Slovenia, and New Zealand, saturate the cumulated number of confirmed case curve rapidly, while this is not the case for countries which did not implement full contact tracing. Furthermore, when intensive school closures and complete contact tracing were implemented at the same time, IRCs rapidly eased. 

Slovenia and New Zealand are seen as typical examples of success: both countries began to implement complete contact tracing before the number of confirmed cases surged and adopted other NPIs in proportion to the increase of confirmed cases. They relaxed Assembly and Movement Restrictions, while keeping full contact tracing, when the increase became stabilized. South Korea is another example of success through implementing full contact tracing throughout the outbreak and maneuvering the stringency of Movement and Assembly Restrictions in response to the number of daily confirmed cases. These approaches were useful not only in containing COVID-19 but also in minimizing government interventions on citizens’ freedom and human rights. On the contrary, the number of confirmed cases continued to grow in the UK, US, and Sweden, where contact tracing was partially implemented or not even implemented for a long time after the first confirmed case. Gradually strengthening Assembly and Movement Restrictions, which were not associated with complete contact tracing, could not help deter the virus transmission in these countries. 

## 4. Discussion

Our analysis shows that school closures must be accompanied by complete contact tracing if they are to be effective in containing COVID-19. All other NPIs, including stay at home requirements and travel bans, are proven to be statistically insignificant or even have a negative impact. However, the fact that NPIs other than school closures are not statistically significant does not mean that they are ineffective in containing COVID-19. The effectiveness of NPIs may vary by country and region depending on the severity of the crisis, the degree of compliance of the people, and the country’s quarantine capacity. Nonetheless, school closures have a statistically significant effect on containing COVID-19 worldwide. This finding is in line with the recent literature on the effectiveness of school closures. Although early studies point out that school closures have little effect on containing COVID-19 [7,25], more recent research presents evidence in favor of the effectiveness of school closing [26,27,28]. 

Furthermore, we can draw four major arguments from our findings. First, state emergency power should be exercised to contain the spread of COVID-19 effectively while minimizing the risk of human rights abuses. One growing concern during the COVID-19 pandemic is that government interventions, aimed at ensuring public safety and protecting citizens’ lives, may result in unnecessarily excessive interference with individual freedom and human rights [4]. The range of human rights at risk during the COVID-19 pandemic is extensive: from rights to life, health, and education to the rights to freedom of movement, public assembly, expression, privacy, and many more. This paper does not assume that one sort of human rights is more critical than another. Concerns have been mainly raised surrounding the perils of contact tracing, in that it may violate individual privacy [23,24]. However, freedom of movement or assembly is as precious as the right to privacy to some people. Considering that all NPIs potentially limit human rights to a certain degree, what a prudent, democratic government should do is to effectively contain COVID-19 while minimally restraining human rights. Our findings imply that implementing full contact tracing and school closure policies, among others, may be a way of achieving these two objectives. Minimizing the risk of human rights abuses should be a principle even when full contact tracing is implemented: any personal information obtained in the process of contact tracing should be handled only for the sole, intended purposes, guaranteeing transparency of the process, and observing domestic and international human rights laws.

Second, when it comes to designing public responses to contain an epidemic that restrains people’s freedom and human rights, a government is required to realize that people’s desire for freedom cannot be easily controlled. Consider the negative correlation between some NPIs, such as workplace closures, public event cancelation, and gathering size restrictions, and DRICs. Restricting larger public events still leads people to having events of smaller sizes. Closing workplaces seldom prevents people from meeting others at cafés and restaurants. Restraining one form of gatherings may push people to take other unforeseeable modes of gathering or mobility, and consequentially, make it even harder for authorities to trace people’s contacts. This implies that designing NPIs requires consideration of how to enhance peoples’ level of compliance. 

Third, we need to think carefully why some NPIs such as travel ban are statistically insignificant. Since the OxCGRT data are national-level data, our findings imply that what is statistically insignificant is some “nationally” implemented NPIs. Restricting people’s movement itself may be effective in containing COVID-19 when implemented in a more targeted way. This means that designing NPIs requires a consideration for the granularity of the result. A travel ban policy does not have to aim at shutting down all domestic or international travel so that the restriction in question is applied to all people and all travels. It could rather purport to seal off infected zones and regulate the travel of people moving to and from the hot spots so that the virus does not spread beyond them. Moreover, the authority could identify vulnerable groups and vulnerable spots and concentrate its limited quarantine capacity to avoid an overburdening of intensive care stations. A travel ban policy targeted at the general public excessively restrains people’s freedom of movement and may not result in enhanced compliance and effective quarantine. 

Fourth, once it is identified that who and what needs to be regulated, full contact tracing needs to be implemented considering at least two measures: comprehensive testing of traced people and treatment of confirmed cases. Considering that around one-third of confirmed cases are asymptomatic [29,30], it is especially important to test all contacts of confirmed people to figure out who was infected and who was not. Then, those confirmed people need to isolate to prevent further transmission and be provided with necessary medical treatment. In other words, contact tracing is a regulatory step identifying the risk of virus transmission, testing is a way of setting up targets of pharmaceutical intervention, and confinement and medical treatment are a means to fix the problem [31,32]. The feasible goal of NPIs may not be obtaining “zero confirmed cases”, but reducing the number of confirmed cases to a level that the health system can handle. 

## 5. Conclusions

Braithwaite pointed out how easily a powerful regulator can be seduced to use coercive options when facing an imminent threat [33]. Several democratic governments acted quickly to employ rather stringent measures such as regional or national lockdowns when facing the first wave of COVID-19 pandemic. It may have been inevitable, because they missed the golden time that the transmission could be regulated through full contact tracing and a targeted approach restricting people’s movement from the initial outbreak. However, when facing the second and third wave of pandemic, governments need to find a balanced way of harnessing state emergency powers that effectively deter the spread of infection while minimally restraining human rights. This should be a goal of any democratic government. It is especially so because NPIs are anticipated to affect our everyday life continuously until vaccines and proper medicines are developed. People will get tired of having to observe their human rights being continuously restrained and their economic lives being threatened. Gaining people’s trust in NPIs is critical to the success of quarantine policies.

Of course this paper has limitations. One of the issues is the reliability of the data. Measuring the strength of NPIs is not easy. OxCGRT [17] provides vast amounts of NPI information. However, the reliability problem of the information still remains, and the standardization method of NPIs causes the scale problem of independent variables. In addition, the number of confirmed cases may not represent the actual number of infected cases. It is mainly dependent upon a government’s testing policy or testing capacity. The, testing policy is especially an essential factor affecting confirmed cases. It differs by country and region, and even in one country it varies depending on the severity of the virus spread. The data used in the current study do not provide such detailed information. 

This study aims to determine how effective NPIs are in containing COVID-19 at a global level. Since each country is the unit of analysis, we employed variables at the macro level, and microscopic factors were not considered. In addition, the seasonal factor could be a significant factor because this study covers countries in northern and southern hemispheres. However, classifying countries into the north and south implies more than the seasonal difference, and the current study has a cross-sectional research design. Therefore, more in-depth longitudinal studies are required to investigate and clarify the worldwide seasonal impact on NPIs for containing COVID-19.

Last but not least, this paper considered a limited number of NPIs that have human rights implications. However, a society’s capacity to fight off the pandemic can be swayed by many factors, such as the quality of the public health care system, government’s risk management capacity and agility, transparent flow of information, political leadership, compliance of businesses and citizens, and many more. Therefore, more extensive studies need to follow to examine what determines the success and failure of containing the COVID-19 pandemic. 

## Figures and Tables

**Figure 1 ijerph-18-00217-f001:**
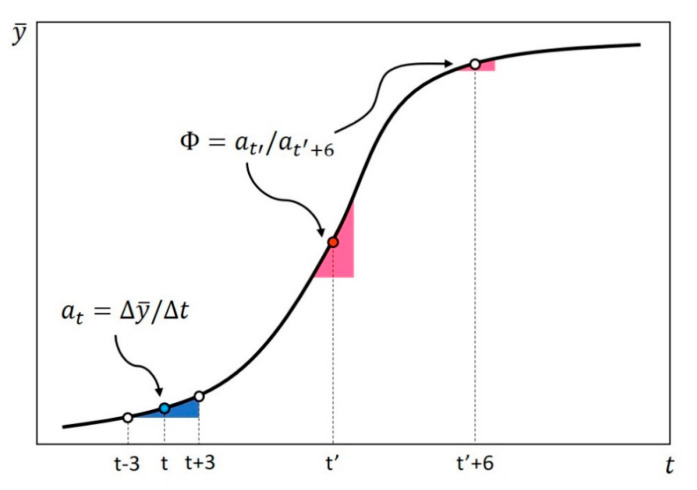
Graphical definition of at and Φ.

**Figure 2 ijerph-18-00217-f002:**
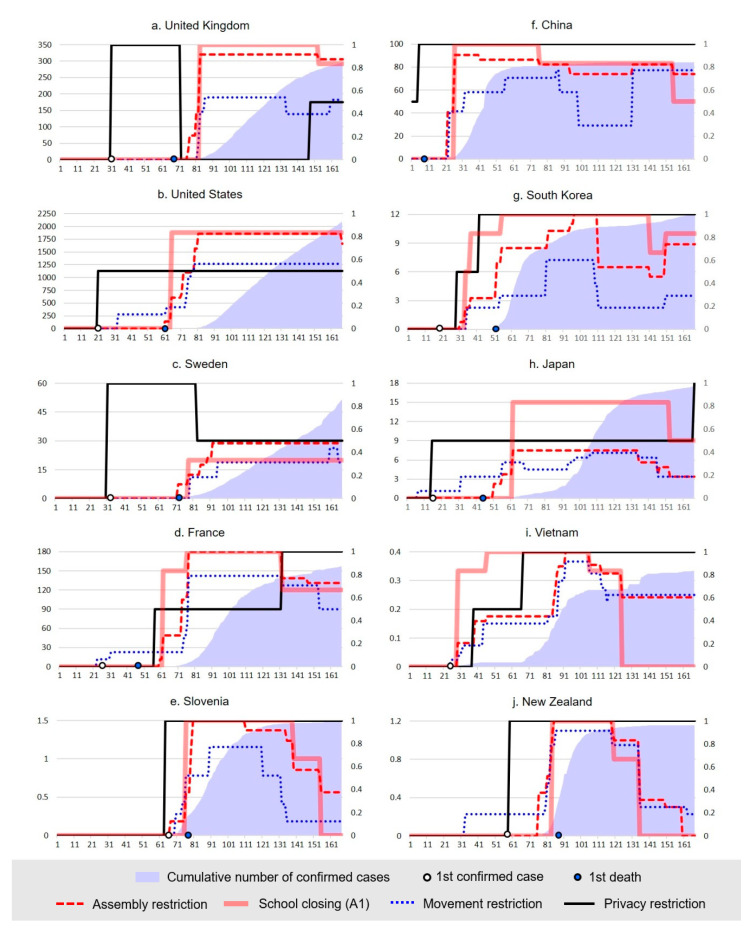
Changes in NPIs and cumulated number of confirmed cases of COVID-19 in the selected countries. The horizontal axis denotes the number of days from the 1 January 2020. The right vertical axis denotes the standardized score of NPIs. The left vertical axis denotes the number of confirmed cases of COVID-19; the unit of confirmed cases is a thousand people.

**Table 1 ijerph-18-00217-t001:** Summary of NPIs (non-pharmaceutical interventions) considered.

Classification	Label	NPIs	Description 1 (Degree)
			Description 2 (Targeted or General)
Assembly Restrictions	A1	School closures	0—No measures; 1—Recommend closing; 2—Require closing (only some levels or categories, e.g., just high school, or just public schools); and 3—Require closing all levels
			0—Targeted; and 1—General
	A2	Workplace closures	0—No measures; 1—Recommend closing (or work from home); 2—Require closing (or work from home) for some sectors or categories of workers; and 3—Require closing (or work from home) for all but essential workplaces (e.g., grocery stores, doctors)
			0—Targeted; and 1—General
	A3	Cancel public events	0—No measures; 1—Recommend canceling; and 2—Require canceling
			0—Targeted; and 1—General
	A4	Restrictions on gathering size	0—No restrictions; 1—Restrictions on very large gatherings (the limit is above 1000 people); 2—Restrictions on gatherings between 101–1000 people; 3—Restrictions on gatherings between 11–100 people; and 4—Restrictions on gatherings of 10 people or less
			0—Targeted; and 1—General
Movement Restrictions	M1	Close public transport	0—No measures; 1—Recommend closing (or significantly reduce volume/route/means of transport available); and 2—Require closing (or prohibit most citizens from using it)
			0—Targeted; and 1—General
	M2	Stay at home requirements	0—No measures; 1—Recommend not leaving the house; 2—Require not leaving the house with exceptions for daily exercise, grocery shopping, and “essential” trips; and 3—Require not leaving the house with minimal exceptions (e.g., allowed to leave only once a week, or only one person can leave at a time, etc.)
			0—Targeted; and 1—General
	M3	Restrictions on internal movement	0—No measures; 1—Recommend not to travel between regions/cities; and 2—internal movement restrictions in place
			0—Targeted; and 1—General
	M4	Restrictions on international travel	0—No measures; 1—Screening; 2—Quarantine arrivals from high-risk regions; 3—Ban on high-risk regions; and 4—Ban on all regions or total border closure
Privacy Restriction	P1	Contact tracing	0—No contact tracing; 1—Limited contact tracing—not done for all cases; and 2—Comprehensive contact tracing—done for all identified cases

Source: OxCGRT [17].

**Table 2 ijerph-18-00217-t002:** Descriptive statistics of the regression variables.

Variables	Mean	Median	SD	Max	Min	N
School closures (A1)	0.931	1.000	0.193	1.000	0.000	108
Workplace closures (A2)	0.671	0.667	0.322	1.000	0.000	108
Cancel public events (A3)	0.904	1.000	0.231	1.000	0.000	108
Gathering size restriction (A4)	0.729	0.875	0.359	1.000	0.000	108
Close public transport (M1)	0.463	0.500	0.376	1.000	0.000	108
Stay at home requirement (M2)	0.457	0.500	0.290	1.000	0.000	108
Internal movement restrictions (M3)	0.631	0.750	0.380	1.000	0.000	108
International travel restrictions (M4)	0.855	1.000	0.258	1.000	0.000	108
Contact tracing (P1)	0.638	0.500	0.353	1.000	0.000	108

**Table 3 ijerph-18-00217-t003:** Correlation analysis of the regression variables.

	DRICs	A1	A2	A3	A4	M1	M2	M3	M4	P1	VIF
DRICs	1.00										-
A1	0.02	1.00									1.55
A2	−0.24 *	0.39 **	1.00								1.67
A3	−0.35 **	0.58 **	0.37 **	1.00							1.88
A4	−0.13 *	0.37 **	0.48 **	0.39 **	1.00						1.39
M1	−0.11	0.36 **	0.44 **	0.37 **	0.38 **	1.00					1.41
M2	−0.26 **	0.41 **	0.58 **	0.47 **	0.43 **	0.49 **	1.00				2.42
M3	−0.32 **	0.38 **	0.45 **	0.44 **	0.41 **	0.49 **	0.71 **	1.00			2.10
M4	−0.21 *	0.10	−0.02	0.36	0.19	0.17	0.09	0.17	1.00		1.17
P1	0.24 *	−0.05	−0.10	−0.16	−0.07	−0.09	−0.07	−0.15	0.04	1.00	1.02

* *p* < 0.05, ** *p* < 0.01.

**Table 4 ijerph-18-00217-t004:** Regression results by models.

Variable	Model 1 (*n* = 108)	Model 2 (*n* = 108)	Model 3 (*n* = 108)
Intercept	2.063 (0.336) ***	1.813 (0.833) *	1.467 (0.884)
School closures (A1)	1.328 (0.382) ***	−2.070 (1.010) *	−1.887 (1.052) ∙
Workplace closures (A2)	−0.434 (0.239) ∙	0.434 (0.459)	0.866 (0.514) ∙
Cancel public events (A3)	−1.226 (0.349) ***	1.568 (0.995)	1.625 (1.004)
Gathering size restrictions (A4)	0.136 (0.195)	0.194 (0.367)	0.178 (0.393)
Close public transport (M1)	0.204 (0.189)	0.209 (0.177)	−0.202 (0.398)
Stay at home requirements (M2)	0.055 (0.318)	0.051 (0.295)	−0.823 (0.650)
Internal movement restrictions (M3)	−0.498 (0.227) *	−0.445 (0.212) *	−0.170 (0.445)
International travel restrictions (M4)	−0.238 (0.248)	−0.308 (0.235)	0.024 (0.418)
Contact tracing (P1)		0.227 (1.033)	0.774 (1.159)
A1:P1		4.162 (1.216) ***	3.930 (1.283) **
A2:P1		−1.221 (0.610) *	−1.771 (0.678) *
A3:P1		−3.163 (1.128) **	−3.245 (1.156) **
A4:P1		−0.053 (0.556)	−0.093 (0.594)
M1:P1			0.627 (0.534)
M2:P1			1.176 (0.832)
M3:P1			−0.436 (0.595)
M4:P1			−0.520 (0.605)
Adjusted R−squared	0.228	0.343	0.346

*** *p* < 0.001, ** *p* < 0.01, * *p* < 0.05, ∙ *p* < 0.1.

## Data Availability

We used the Oxford COVID-19 Government Response Tracker (OxCGRT) database for confirmed case counts for the 108 countries and changes in their NPIs for the time period in our study, and this is available at https://www.bsg.ox.ac.uk/research/research-projects/coronavirus-government-response-tracker#data.

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
