# Peer review of "Effect of COVID-19 Non-Pharmaceutical Interventions and the Implications for Human Rights"

_ijerph, 2020, doi:10.3390/ijerph18010217_

Round 1

Reviewer 1 Report

The paper covers an interesting topic, but also leaves a number of questions open.

First, it does not engage with literature pointing us in another direction (many see school closings as little effective, especially if done without age differentiation).

Second, the result seems to be that contact tracing is most effective and measures have to be taken to allow this contact tracing. But as you also classify it as the most problematic HR restriction, is`it then the measure of choice and more "controllable" settings have to me created? 

Third, decision makers had a clear goal with all these measures: protect particularly vulnerable groups and get incidences down quickly, mainly to avoid an overburdening of intensive care stations; the measures are not related to the specific goals of policy-makers; 

Author Response

Thank you very much for your careful reading and insightful comments on our manuscript. We are happy to see that you are overall positive about the manuscript. We have sincerely attempted to address all of them. Please find the attached word document for detailed responses.

Reviewer 2 Report

Effective control of Covid-19 infection is a major public health concern today, and the subject matter of this article fits fairly with the scope of the journal. The fact that the article has been reviewed from an international perspective, including many regions, is also commendable. In particular, ELSI in the implementation of social distansing is an extremely important issue not only for the protection of human rights, but also for the implementation of effective infectious disease control.
On the other hand, in order to argue for the invalidity of these social distancing policies in this study, a more statistically valid methodology and a more rigorous examination of the analytical process are required.

Firstly, many problems can be identified with the methodologies used to analyse the data; the items and measures based on the OxCGRT are shown in Table 1. Whilst the items are comprehensive, there are statistical problems in setting up the scales. Although the authors treated all of them as ratio scales, many of them are rank or interval scales, making them difficult to apply to regression analysis; There is also no mention of normality, linearity or independence of each item; No mention of multicollinearity between items., etc. The authors first require an extensive review of the key points in the design of the analysis before jumping to conclusions.

Secondly, the authors have adopted an index based on the cumulated number of confirmed cases as an explanatory variable in the linear regression model. This choice is not necessarily appropriate given the differences in testing policies in different countries (in extreme cases, whether to test all citizens including asymptomatic citizens or only those with symptoms) and the possibility of arbitrary changes in these policies. Measures need to be taken to design indicators that take these realities and possibilities into account, or leave room for debate in Discussion.

Thirdly, differences amongst regions will need to be taken into account. In particular, countries in the Northern vs. Southern Hemisphere differs significantly in terms of seasonal factors over the same observed period; The stage at which strong social isolation, including lockdowns, was implemented is also an important point to examine; The timing of onset of infection, the route of transmission and the predominant genotype of COVID-19 viruses will also differ amongst countries/regions. Although it will be difficult to take into account all of these factors in detail, representative factors should be reflected in the analysis and others should be addressed in Discussion.

The following are minor comments:

  • A description of the study's findings and conclusions can be found in Chapter 1 and should be deleted.
  • A review of cases of acute infection prior to COVID-19 is needed.
  • Interaction terms were introduced into the models 2 and 3, however, there needs to be an examination of the hypothesis on which they were based.
  • Typo in L196 'be2' that leads a need for further consideration of word editing to the entire text.

Author Response

(The authors gave the same response as above.)

Reviewer 3 Report

The paper presents a study to assess which NPI against COVID-19 are the most effective ones. I don't have any problem with respect to the results, and I trust the authors is doing a scientifically robust regression analysis.

My problem with this paper is that the analysis is covered by an envelope talking about human rights. While it is an interesting topic, and it could be discussed in the discussion section, the mathematical model and the data use do not provide any insights about how people feel about their rights being decreased. Therefore I encourage to drop the discussion about human rights and limit the paper to only analyzing which measures have been more effective. For the authors to include a discussion on human rights, they would have to widen the scope of their research design. Incidentally, I find strange that the authors end up recommending the type of measures to which people appear to be most against, due to privacy concerns.

A second point of concern is the overrepresentation in their data sample of 'bad cases'. There are very few countries in the world which have managed to control the pandemic, and it remains unclear whether the type of NPI adopted in those few countries can be generalized to others (due to cultural, monetary and logistic issues).

Author Response

(The authors gave the same response as above.)

Round 2

Reviewer 2 Report

I have read the revised manuscript and the correspondence and confirm that the authors have appropriately addressed the points suggested. I agree with the acceptance of the manuscript.

I believe the present paper brings significant findings. Congratulations on your publication.

Reviewer 3 Report

The authors have tackled my comments appropriately